# TRPV1 Contributes to Modulate the Nitric Oxide Pathway and Oxidative Stress in the Isolated and Perfused Rat Heart during Ischemia and Reperfusion

**DOI:** 10.3390/molecules27031031

**Published:** 2022-02-03

**Authors:** Vicente Castrejón-Téllez, Leonardo del Valle-Mondragón, Israel Pérez-Torres, Verónica Guarner-Lans, Gustavo Pastelín-Hernández, Angélica Ruiz-Ramírez, Julieta Anabell Díaz-Juárez, Elvira Varela-López, Víctor Hugo Oidor-Chan, Alvaro Vargas-González, Raúl Martínez-Memije, Pedro Flores-Chávez, Bruno León-Ruíz, Sergio Arriaga-Carrillo, Juan Carlos Torres-Narváez

**Affiliations:** 1Departamento de Fisiología, Ciudad de México 14080, Mexico; vcastrejn@yahoo.com.mx (V.C.-T.); gualanv@yahoo.com (V.G.-L.); alvaro.vargas@cardiologia.org.mx (A.V.-G.); 2Departamento de Farmacología Dr. Rafael Méndez Martínez, Ciudad de México 14080, Mexico; leonardodvm65@hotmail.com (L.d.V.-M.); pastelingustavo@gmail.com (G.P.-H.); anabelldij@gmail.com (J.A.D.-J.); victorhugooidor@gmail.com (V.H.O.-C.); 3Departamento de Biomedicina Cardiovascular, Ciudad de México 14080, Mexico; pertorisr@yahoo.com.mx (I.P.-T.); angelica_ruiz_ramirez@yahoo.com.mx (A.R.-R.); 4Laboratorio de Cardiología Translacional, Unidad de Investigación en Medicina Translacional UNAM/INCAR, Ciudad de México 14080, Mexico; varelopz@yahoo.com; 5Departamento de Instrumentación Electromecánica, Instituto Nacional de Cardiología Ignacio Chávez, Ciudad de México 14080, Mexico; raulmmemije@yahoo.com (R.M.-M.); pelfoch07@yahoo.com.mx (P.F.-C.); 6Departamento de Atención a la Salud, Universidad Autónoma Metropolitana Unidad Xochimilco, Ciudad de México 04960, Mexico; brarchnmy@hotmail.com (B.L.-R.); sarriaga24@gmail.com (S.A.-C.)

**Keywords:** ischemia reperfusion, TRPV1, cardioprotection

## Abstract

The transient vanilloid receptor potential type 1 (TRPV1) regulates neuronal and vascular functions mediated by nitric oxide (NO) and by the calcitonin gene-related peptide (CGRP). Here, we study the participation of TRPV1 in the regulation of myocardial injury caused by ischemia-reperfusion and in the control of NO, tetrahydrobiopterin (BH4), the cGMP pathway, CGRP, total antioxidant capacity (TAC), malondialdehyde (MDA) and phosphodiesterase-3 (PDE-3). Isolated hearts of Wistar rats perfused according to the Langendorff technique were used to study the effects of an agonist of TRPV1, capsaicin (CS), an antagonist, capsazepine (CZ), and their combination CZ+CS. The hearts were subjected to three conditions: (1) control, (2) ischemia and (3) ischemia-reperfusion. We determined cardiac mechanical activity and the levels of NO, cGMP, BH4, CGRP, TAC, MDA and PDE-3 in ventricular tissue after administration of CS, CZ and CZ+CS. Western blots were used to study the expressions of eNOS, iNOS and phosphorylated NOS (pNOS). Structural changes were determined by histological evaluation. CS prevented damage caused by ischemia-reperfusion by improving cardiac mechanical activity and elevating the levels of NO, cGMP, BH4, TAC and CGRP. TRPV1 and iNOS expression were increased under ischemic conditions, while eNOS and pNOS were not modified. We conclude that the activation of TRPV1 constitutes a therapeutic possibility to counteract the damage caused by ischemia and reperfusion by regulating the NO pathway through CGRP.

## 1. Introduction

Cardiovascular diseases are a serious health problem worldwide causing significant mortality. Risk factors for cardiovascular disease include unhealthy nutrition, physical inactivity, dyslipidemia, hyperglycemia, high blood pressure, obesity, thrombosis, smoking and kidney dysfunction. Job stress in large cities and genetic load also cause or magnify the risk factors for cardiovascular diseases [1,2,3]. All these conditions also lead to diabetes mellitus, heart failure, atherosclerosis, hypertension and generalized damage to organs and tissues resulting in an increased prevalence of acute myocardial infarction (AMI). During AMI, blood flow to the heart is blocked, damaging cardiomyocites.

When perfusion is restored after AMI, damage by reperfusion results, consisting of the opening of the mitochondrial permeability transition pore, an increase in oxidative stress, cytosine activation, lipid peroxidation, and cell death due to necrosis or apoptosis. Moreover, pathways such as the injury salvage kinase (RISK) and the survivor activating factor enhancement (SAFE) are activated [4,5,6,7,8,9].

The Transient Receptor Potential Vanilloid type1 (TRPV1) is a non-selective cation channel that allows the passage of H^+^, Na^+^, Ca^2+^ y Mg^2+^ ions [10]. Although it was initially identified on sensory nerve fibers, it has been found having a high expression in other tissues and cells including trigeminal ganglia, dorsal root ganglia, neurons, urinary bladder, testis, adipocytes, smooth muscle cells, endothelial cells, pancreatic β-cells, liver, heart, skeletal muscle, and kidney cells. In the cardiovascular system, TRPV1 is expressed in the ventricles, endothelial cells, vascular smooth muscle cells, and on sensory neurons innervating the myocardium [11]. TRPV1 is activated by stimuli such as shear stress, changes in pH or temperature in the vasculature. Endogenously, these receptors are also activated by anti-inflammatory responses and by anandamide. Anandamide is a derivative of arachidonic acid metabolism and also activates type 1 cannabinoid receptors (CB1). In experimental research, agonists such as capsaicin (CS) which is the active principle in plants of the *Capsicum* genus and antagonists such as capsazepine (CZ) are frequently employed to study the actions of the TRPV1 receptors [12,13,14].

In the last two decades, TRPV1 has been proposed as a therapeutic possibility to reduce the effects of pain-related diseases such as migraine and other diseases including rheumatoid arthritis [15,16,17]. Moreover, CS and neuronal depletion of TRPV1 have been used as strategies in pain sensation studies. Although, inactivation of non-neuronal TRPV1 with high doses of CS has not yet been observed, this topic requires further studies [18]. Since TRPV1 participates in the regulation of Ca^2+^ flux through the cell membrane and regulates the synthesis of NO, its role in some pathologies such as arterial hypertension, diabetes mellitus and ischemia-reperfusion injury are currently being studied [11,14,18,19]. When TRPV1 is activated, neuropeptides such as substance P and the calcitonin gene-related peptide (CGRP) are discharged from the perivascular sensory nerves that innervate the myocardium to exert its cardioprotective effects [19,20].

The purpose of this work was to demonstrate that the activation of TRPV1 at a systemic level generates cardioprotection, preventing the damage caused by ischemia and reperfusion. TRPV1 might also control oxidative stress and therefore, total antioxidant capacity (TAC) was measured [20,21]. We also analyzed the levels of some cell damage biomarkers that are overexpressed by biological processes related to NO deficiency in the heart, such as malondialdehyde (MDA) (the end product of lipoperoxidation) and phosphodiesterase-3 (PDE-3). Once activated, PDE-3 induces the degradation of cGMP to guanosine monophosphate (GMP), inhibiting vascular smooth muscle relaxation [22].

The biomarkers studied in this paper are overexpressed or inhibited by various biological processes related to oxidative stress and cell damage. This is the case for MDA, which increases cell damage and death, and PDE-3, which alters the pathway of cGMP, inhibiting relaxation. CGRP, BH4, NO and cGMP are indicators of alterations in vasorelaxation. These biomarkers are sensitive to changes in the vascular environment or to diseases. Moreover, previous studies suggest that TRPV1 is a regulator of the cGMP-NO pathway.

Our hypothesis was that treatment with CS would activate TRPV1 in the heart under conditions of ischemia and reperfusion, restoring control of Ca^2+^ flow into cells and elevating the bioavailability of NO, thereby reducing ROS and enhancing the cardioprotective mechanisms.

## 2. Methods

Male Wistar rats of 300–350 g were used. They were provided by the Laboratory Animal Care of the National Institute of Cardiology “Ignacio Chávez” in México. All procedures for handling animals were approved by the Institutional Ethics Committee and in accordance with the National Rules for the care and handling of experimental animals (SAGARPA, NOM-062-ZOO-1999). The animals were kept under optimal conditions of temperature and light (12 h light/dark) with a standard diet (Lab diet 5012, PMI Nutrition International, Richmond, IN, USA) and water ad libitum. The experimental animals were randomly grouped as follows: 1.—Control; 2.—CS treatment (final dose of 20 mg/kg which was distributed over 4 days); 3.—CZ treatment (final dose of 24 mg/kg distributed over 4 days) and 4.- CZ+CS treatment for 4 days. The CZ was applied first and, an hour later, the CS was injected. CS was applied at a lower dose (20 mg/kg) than that reported by Zhou et al. and by Szabados et al. [18,23] to avoid deletion of nerve endings and also because we performed a curve dose response in previous experiments with CS and CZ in isolated hearts [24]. The application of the drugs was through a s.c. injection.

### 2.1. Reagents

The reagents used in this work were analytical grade (Sigma Chemical Co., St. Louis, MO, USA). They included Capsaicin (8-methyl-N-vanillyl-6-nonenamide) agonist of TRPV1; Capsazepine [N-(2-(4-chlorophenyl)ethyl)-1,3,4,5-tetrahydro-7,8-dihydroxy-2H- 2-benzazepine-2-arbothioamide] antagonist of CS. CS and CZ were diluted separately in ethanol-water 2:1.

### 2.2. Determinations in the Isolated and Perfused Heart According to the Langendorff Technique

The experimental animals were anesthetized with sodium pentobarbital (60 mg/Kg of body weight) and heparin (1000 U/mL/Kg of body weight) was administered. Respiratory assistance was applied by means of a Palmer air pump and a cannula in the trachea inserted through a small incision.

The heart was exposed by a thoracotomy and removed. It was then placed in ice-cold Tyrode solution to be arrested and to prevent ischemic preconditioning. Immediately, the heart was connected to the perfusion system through the ascending aorta. The heart was maintained with mechanical activity with Krebs–Henseleit solution with the following composition (mM): 120 NaCl, 23.4 NaHCO_3_, 4.8 KCl, 1.2 KH_2_PO_4_, 0.86 MgSO_4_, 1.25 CaCl_2_ and 11.0 of glucose (pH 7.4 and temperature 37 °C) through a constant retrograde perfusion (13 mL/min). The perfusion started with an adaptation period of 30 min (5 min with a flow (F) of 25 mL/min and 25 min with F of 13 mL/min). Heart rate (HR) was maintained at 312-324 beats per minute, using a Grass stimulator (U7, Grass Instruments Co., Quincy, MA, USA).

Coronary flow was regulated with a peristaltic pump (SAD22, Grass Instruments Co., Quincy, MA, USA). Parameters including left intraventricular pressure (LIVP) were recorded by means of a Grass hydropneumatic pressure transducer, to which a catheter with a latex balloon was connected. The balloon was introduced through the mitral valve into the left ventricle and once inside the cavity, an internal pressure of 5–10 mmHg (diastolic pressure) was applied.

With another Grass hydropneumatic pressure transducer, the perfusion pressure (PP) was recorded and a range of 55–70 mmHg at the beginning of the experiment was considered as an inclusion criterion.

All parameters were recorded using a computer acquisition data system (Grass Poly View). Cardiac mechanical activity (CMA) was calculated as HR × LIVP = CMA [4,25]. The experimental conditions are illustrated in Figure 1.

### 2.3. Determination of NO, BH4, TAC, cGMP

#### 2.3.1. Sample Treatment

Ventricular tissue samples for determination of nitric oxide (NO), tetrahydrobiopterin (BH4), and total antioxidant capacity (TAC), were homogenized in 5mM phosphate buffer at pH 7.4 and were centrifuged at 16,000× *g* for 15 min at 10 °C. (Sorvall SR70, Thermo Scientific Inc., Urbana, IL, USA). Centrifugation was carried out at 10 °C because these biomolecules remain stable below 18 °C. We selected an intermediate temperature to facilitate the formation of the button. The supernatants were filtered using a 0.22 µm nitrocellulose filter (Millipore, Billerica, MA, USA), and they were reserved at 4–8 °C for later analysis.

For the determination of cGMP, the ventricular tissue was homogenized in a 100 mM phosphate buffer at 4 °C and a pH of 7.5 ± 0.05. Centrifugation (Sorvall SR70, Thermo Scientific Inc., Urbana, IL, USA) was performed at 16,000× *g* for 15 min at 10 °C. The supernatant was diluted 1:10 with 0.05M NaOH and filtered with 0.22 µm filters (Millipore Billerica, MA, USA). It was then reserved at 4–8 °C for later analysis.

#### 2.3.2. Sample Analysis

##### Nitric Oxide

The determination was made directly on the tissue homogenate by a spectrophotometric method in the UV-Vis region (490 nm), using a DW2000 spectrophotometer (SLM-Aminco, SLM Instruments Inc., Urbana, IL, USA) in the supernatant obtained in point 2.3.1. The equipment was adjusted to zero absorbance with a reagent blank. NO concentration was determined using a standard curve from 0–500 pmol/mL using analytical grade sodium nitrite [26].

##### Total Antioxidant Capacity

The determination was made directly in the supernatant obtained in point 2.3.1. A spectrophotometric method was used for TAC quantification. Samples were analyzed in UV-Vis region at a room temperature (Cary 4000, Varian Inc., Mulgrave, VIC, Australia) at 720 nm. The equipment was adjusted to zero fluorescence with a reagent blank. TAC concentration was determined using a standard curve from 0–1500 µmol/L using analytical grade ferrous sulfate [27].

##### Tetrahydrobiopterin

The supernatant obtained in point 2.3.1. was diluted 1:10 with 0.01M NaOH before starting the analysis. The determination of BH4 levels in ventricular tissue was performed by capillary zone electrophoresis with UV-Vis detection by diode array, using the methodology of Han [28].

The sample was passed through a Sep-Pak Classic C18-NH2 cartridge (Waters, Urbana, IL, USA) and directly analyzed with a P/ACETM MDQ (Beckman Coulter Inc., Fullerton, CA, USA) at 30 kV for 6 min and a wavelength of 230 nm at 10 °C, using a running buffer (0.1 M Tris—0.1 M boric acid—2 mM EDTA, pH 8.75). The samples were injected under hydropneumatic pressure of 0.5 psi/10 s. BH4 concentration was determined using a standard curve from 0–200 pmol/mL using analytical grade BH4.

##### Cyclic Guanosin Monophosphate

The evaluation of cGMP levels in the ventricular tissue sample obtained in point 2.3.1 was performed by capillary zone electrophoresis (P/ACETM MDQ system; Beckman Coulter Inc., Fullerton, CA, USA) [29]. The filtrated ventricular tissue homogenate was deproteinized with cold methanol in a 1:1 ratio. Centrifugation was performed at 16,000× *g* for 10 min (Sorvall SR70, Thermo Scientific Inc., Urbana, IL, USA) and the supernatant was diluted 1:10 with 0.05M hydrochloric acid and filtered with a 0.22 µm nitrocellulose membrane (Millipore Billerica, MA, USA) and directly analyzed. The separation was carried out at −25 kV for 15 min at 190 nm using a running buffer (40 mM citric acid + 0.8 mM CTAB at pH 4.4). cGMP concentration was determined using a standard curve from 0-200 pmol/mL using analytical grade cGMP.

### 2.4. Determination of Calcitonin Gene Related Peptide

The CGRP analysis was carried out according to the Seon et al. method. The processed samples were purified by preparative reverse-phase HPLC (ACQUITY UPLC System, Waters Corporation, Barcelona, España) on a Waters RCM compact preparative cartridge Delta-Pak C 18 (300 Å; 25 3 100 mm/Waters, Barcelona, España) eluted at a flow rate of 8 mL/min by a linear gradient of acetonitrile in 0.1% trifluoroacetic acid in water (5 min; wash at 5% acetonitrile followed by a 5–60% linear gradient of acetonitrile at 0.5%/min). The purified sample was subjected to a HPLC analysis (ACQUITYUPLC System, Waters Corporation, Barcelona, España), using Lichrosorb C18 column (5 mm, 4.6 × 250 mm/Phenomenex, CA, USA), and lineal acetonitrile gradient (20–58%) at a flow rate of 0.75 mL/min, and detection to 220 nm at 10 °C. The concentration of the CGRP was determined by a standard curve of 0–100 fmol/mL [30].

### 2.5. Damage Indicators Determinations (PDE-3 and MDA)

#### 2.5.1. Phosphodiesterase 3

To quantify PDE-3 levels in ventricular tissue [31], the tissue was homogenized in cold buffer with 100 mM saccharose+20 mM HEPES + 50 mM citrates at pH 5.6. Centrifugation was performed at 16,000× *g* for 15 min at 10 °C (Sorvall SR70 centrifuge, Thermo Scientific Inc., Urbana, IL, USA). The supernatants were filtered with 22-µm nitrocellulose filters (Millipore, Billerica, MA, USA); and the analysis was carried out directly. All samples were stored at 70 °C until the day of analysis.

The analysis of PDE-3 levels was carried out in ventricular tissue by capillary zone electrophoresis, using UV-Vis detection by diode array. The filtrate was deproteinized with cold methanol, cold trichloroacetic acid in a ratio of 10:1. Centrifugation was performed at 16,000× *g* for 15 min at 10 °C (Sorvall SR70, Thermo Scientific Inc., Urbana, IL, USA). The filtration was carried out with a 0.22 µm nitrocellulose membrane (Millipore Billerica, MA, USA). It was subsequently diluted 1:10 with cold 0.1 M sodium hydroxide. Sep-Pak Classic C-18 cartridge (Waters, Urbana, IL, USA).

The sample was passed through a 100 mM citrate buffer and pH 2.5. The analysis was performed directly with P/ACETM MDQ System (Beckman Coulter Inc., Fullerton, CA, USA). The separation was carried out at 20 kV for 30 min at 240 nm at 20 °C and a running buffer (100 mM boric acid at pH 2.8). Hydrodynamic pressure 0.5 psi/10s. The concentration of the PDE-3 was determined by a standard curve of 0–100 pmol/mL.

#### 2.5.2. Malondialdehyde

The supernatant was diluted 1:10 with 0.1 M NaOH before starting the analysis. MDA was determined in ventricular tissue by capillary zone electrophoresis (P/ACETM MDQ System; Beckman Coulter Inc., Fullerton, CA, USA). The separation was carried out at −20 kV for 4 min at a wavelength of 267 nm at 10 °C using a 100mM borate buffer +0.5 mM CTAB at pH 9.0. The samples were injected at a hydropneumatic pressure of 0.5 psi/10 s. The concentration of the MDA was determined by a standard curve of 0–200 pmol/mL [32].

### 2.6. eNOS, pNOS, iNOS and TRPV1 Detection

The ventricular tissue was homogenized in a mortar with liquid nitrogen and incubated 1 h in lysis buffer (50 mM Tris pH 8.0, 150 mM NaCl, 15 mM Imidazole, 10% Glycerol, 100 mg/10 mL DOC, 1% Triton X-100) supplemented with a mixture of protease inhibitors (leupeptin, aprotinin, pepstatin, PMSF) (Sigma Chemical Co., St. Louis, MO, USA). Protein was determined by the Bradford method (Bio-Rad laboratories) [33]. A total of 50 μg of the ventricular tissue homogenate were mixed with 3X loading buffer (20% glycerol, 4% SDS, 0.02% bromophenol blue, 0.2% 2-mercaptoethanol, 125 mM Tris, pH 6.8) and heated for 5 min at 100 °C. SDS-PAGE to 8% was used to separate the proteins eNOS, pNOS, iNOS and TRPV1. Then, the proteins were transferred to PVDF membrane (0.22 μm). The blots were blocked for 1h at room temperature using the Tris saline buffer, plus 0.01% Tween (TBS-T) and 5% non-fat dehydrated milk. Afterwards, membranes were incubated with primary antibodies overnight in a 1:1000 dilution, at 4 °C. The primary antibodies used were mouse anti-e-NOS, mouse anti-p-NOS, mouse i-NOS (all from Santa Cruz Biotechnology, Santa Cruz, CA, USA) and rabbit anti-TRPV1 (from Sigma-Aldrich, St. Louis, MO, USA). TBS-T buffer was used to rinse the membranes four times and then membranes were incubated overnight at 4 °C with horseradish peroxidase conjugated secondary antibodies at a dilution of 1:10,000 (Santa Cruz Biotechnology, Santa Cruz, CA, USA). All blots were incubated as a control with the β-Actin antibody (sc-32251) (Santa Cruz Biotechnology, Santa Cruz, CA, USA). A chemiluminescence assay (Clarity Western ECL Substrate, Bio-Rad Laboratories, Inc., Hercules, CA, USA) was used for protein detection. X-ray films (AGFA, Ortho CP-GU, Agfa HealthCare NV, Mortsel, Belgium) were used to detect the emitted chemiluminescence. A GS-800 densitometer (including Quantity One software from Bio-Rad Laboratories, Inc.) was employed to acquire the images from each film. We expressed the values of each band density as arbitrary units (AU).

### 2.7. Histological Sections

Another group of animals under the same experimental conditions had their hearts extracted to make histological cross sections which included the left and right ventricular chambers. The sections were washed in a 0.9% NaCl solution and fixed in 10% formaldehyde and pH 7.4. Once the tissue was perfectly fixed, it was processed according to conventional histological procedures and stained by Masson’s trichrome at 16×. All representative microphotographs of the groups were taken from areas irrigated by the left anterior descending coronary artery, from the tip of the heart and from the anterior wall of the left ventricle, and approximately two-thirds anterior to the ventricular septum. Observations were performed at 16×.

Analysis of the marked histological sections was carried out with a Carl Zeiss light microscope (Carl Zeiss Axio Imager Z2, West Germany objective EC Plan-Neofluar 16×) and with an HP Z800 computer and an HP ZR30W screen. The photomicrographs were analyzed by densitometry using the SigmaScan Pro 5 Image Analysis software.

## 3. Statistical Analysis

Results are expressed as mean ± standard error of the mean (SEM). Differences were considered statistically significant when *p* ≤ 0.05. The different symbols in figures indicate significant differences. We applied a one-way analysis of variance (ANOVA) followed by a Tukey post hoc test using the SigmaPlot program version 11 (Jandel Scientific, San Jose, CA, USA).

## 4. Results

### 4.1. Cardiac Mechanical Activity

The Langendorff’s model of the isolated heart allowed us to study the mechanical activity of the heart under our three experimental conditions: normal perfusion, ischemia and ischemia+reperfusion. Figure 2 shows that the cardiac mechanical activity of the group without ischemia and reperfusion remained constant during the 120 min that lasted the experiment (approximately 23,000 mmHg/beats/min^−1^ (black circles)). This curve was compared to that of another group of hearts in which a global ischemia of 30 min was induced and reperfusion was allowed for 60 min (open circles). During reperfusion, the mechanical function of the heart was restored to 80%, but it did not reach 100%.

In the CS treated group, the heart completely recovered its normal mechanical activity during reperfusion (open squares). The action of the CS treatment was compared with that from a group of hearts from rats treated with CZ and from the group of rats treated with CZ+CS. In both of these groups, cardiac mechanical activity was decreased since the period prior to ischemia, and it remained low during reperfusion, especially in the CZ+CS group.

### 4.2. NO, BH4 and cGMP

To explain the changes in cardiac work produced by the treatments during ischemia and reperfusion, we evaluated important factors that participate in the NO pathway, such as NO, BH4, and cGMP levels in ventricular tissue samples. Figure 3A shows that there were higher levels of NO in the hearts of animals treated with CS that were not subjected to ischemia (7.6 ± 0.9 pmol/mg of protein) with respect to the control (3.3 ± 0.52 pmol/mg of protein). CS actions upon this variable were significantly inhibited by CZ to 1.3 ± 0.15 and by CZ+CS to 1.4 ± 0.2 pmol/mg of protein.

In hearts with global ischemia for 30 min CS maintained NO values without significant changes with respect to the control hearts (2.8 ± 0.6 and 1.8 ± 0.8 pmol/mg of protein). Treatments with CZ and CZ+CS decreased the action of CS (to 0.3 ± 0.09 and to 0.9 ± 0.13 pmol/mg of protein, respectively).

During reperfusion, CS maintained NO at the same level as in control hearts (3.6 ± 1.2 and 3 ± 0.17 pmol/mg of protein). These levels were decreased by CZ (to 0.4 ± 0.02 pmol/mg of protein). The CZ+CS combination did not induce significant changes.

We measured the BH4 levels in the hearts. BH4 is an essential cofactor for NO synthesis due to its molecular coupling with eNOS. Figure 3B shows that the BH4 levels are modified in a similar way to those of NO (panel A) (3.8 ± 0.7 pmol/mg of protein). The BH4 levels were increased (8.7 ± 0.7 pmol/mg of protein) in the hearts of the CS group that were perfused for 1 h. This effect was diminished by CZ (to 1.7 ± 0.2 pmol/mg of protein) and by CZ+CS (to 1.8 ± 0.3).

The BH4 levels in the group with CS were of 3.1 ± 0.4 pmol/mg of protein in the period of global ischemia. They were significantly decreased in the groups treated with CZ and with CZ+CS (to 0.6 ± 0.1 and to 0.8 ± 0.1 pmol/mg of protein, respectively). During reperfusion, the levels of BH4 were significantly decreased by CZ (to 0.8 ± 0.1) and with CZ+CS (to 1.8 ± 0.3 pmol/mg of protein), with respect to the group with CS (8.6 ± 2 pmol/mg of protein).

cGMP plays a preponderant role for vessel relaxation when the NO pathway is activated. Figure 3C shows that the levels of cGMP in the control hearts during perfusion and ischemia did not change significantly; however, there was a tendency to increase in the control hearts during reperfusion. The effect of CS was inhibited by CZ and by CZ+CS (to 0.192 ± 0.03 and to 0.196 ± 0.009 pmol/mg of protein, respectively).

During ischemia, cGMP levels in the heart decreased with CZ (to 0.051 ± 0.017) and with CZ+CS (to 0.22 ± 0.038 pmol/mg of protein, respectively) with respect to the group with CS (0.483 ± 0.103 pmol/mg of protein). During reperfusion, cGMP levels decreased with CZ (to 0.076 ± 0.004 pmol/mg of protein) and with CZ+CS (to 0.22 ± 0.038), with respect to CS group (1.337 ± 0.42 pmol/mg of protein).

### 4.3. TAC and CGRP

The activation of TRPV1 was reflected in an increase in the TAC and CGRP levels (Figure 4). Figure 4A shows that, during the perfusion period, the TAC was significantly increased by the treatment with CS when compared to the control (0.021 ± 0.004 to 0.041 ± 0.003 mmol/mg of protein). This increase was inhibited by CZ (to 0.008 ± 0.0011) and by CZ+CS (to 0.005 ± 0.004 mmol/mg of protein).

In the ischemic period, the TAC showed no significant differences with respect to the control of the perfusion period in the group without treatment and in the group with CS. The CZ and by CZ+CS treatments significantly decreased its level (to 0.002 ± 0.0005 and 0.008 ± 0.0013 mmol/mg protein, respectively). A similar effect was observed during reperfusion since the effect of CS (0.051 ± 0.012) was inhibited by CZ and by CZ+CS (0.003 ± 0.0001 and 0.008 ± 0.0013 mmol/mg of protein, respectively).

CGRP is a neuropeptide that has recently been related to the vasodilator actions of NO. Figure 4B shows that CS significantly increased the levels of CGRP in the perfusion period in comparison with the control (0.012 ± 0.002 to 0.042 ± 0.004). This effect was inhibited by CZ (0.004 ± 0.001) and by CZ+CS (0.003 ± 0.001 fmol/mg of protein). During ischemia, the CS kept the CGRP level very close to the baseline value (0.013 ± 0.001). This level was decreased by CZ (to 0.001 ± 0) and by CZ+CS (to 0.005 ± 0.001 fmol/mg of protein).

During reperfusion, there was an important and significant increase in CGRP in the CS group with respect to the control (0.009 ± 0.002 to 0.04 ± 0.007 fmol/mg of protein), taking into account that the damage due to ischemia and reperfusion had already occurred. CS-stimulated CGRP production was inhibited by CZ (to 0.003 ± 0.001) and CZ+CS (to 0.003 ± 0.001 fmol/mg of protein).

### 4.4. MDA and PDE-3 Levels

The activation of TRPV1 was reflected by alterations in the levels of MDA and PDE-3. The levels of MDA and PDE-3 with the CS treatment remained the same as in their controls during perfusion, ischemia and reperfusion. In the perfusion period, PDE-3 was significantly increased in the hearts of rats treated with CZ (to 0.173 ± 0.013), but a high level was maintained with CZ+CS (to 0.086 ± 0.007 pmol/mg of protein), with respect to the CS group (0.13 ± 0.007).

During ischemia, PDE-3 levels were increased when compared to hearts in the perfusion period without treatment (0.025 ± 0.003 to 0.117 ± 0.013 pmol/mg of protein). CS regulated PDE-3 levels were maintained at 0.055 ± 0.016 pmol/mg of protein. The PDE-3 increased with CZ (to 0.173 ± 0.013) and with CZ+CS (to 0.193 ± 0.021 pmol/mg of protein) in the CS group. With the restoration of coronary flow, there was a tendency to re-establish PDE-3 baseline values, except with the CZ+CS treatment, with which the level remained significantly elevated when compared to the CS group.

### 4.5. Lipoperoxidation

The lipoperoxidation biomarker MDA maintained the same levels during the perfusion period in the control (0.002 ± 0.0001) and in the CS group (0.0015 ± 0.0001). MDA significantly increased with CZ (to 0.0.0112 ± 0.003) and with CZ+CS (to 0.0087 ± 0.002 pmol/mg of protein). In the ischemic period, CS maintained a low level of MDA (0.0056 ± 0.001) when compared to its control (0.0087 ± 0.0025) and the level of MDA decreased in the CZ treatment (0.0219 ± 0.003) and CZ+CS treatment (0.027 ± 0.005 pmol/mg of protein).

This same tendency was observed in reperfusion. In contrast, treatments with CZ and with CZ+CS raised MDA levels (to 0.021 ± 0.002 and to 0.011 ± 0.002 pmol/mg of protein) in comparison with the CS group (0.002 ± 0.0004).

### 4.6. Nitric Oxide Synthase

Figure 5 shows that there are no important modifications in the expression of total eNOS under any of the experimental conditions. Panel A shows the relative density of eNOS in control hearts with perfusion for 30 and 120 min, ischemic hearts, and hearts with ischemia and reperfusion. Panels B, C and D also show the relative density of eNOS under the same conditions as in panel A, but in hearts of rats treated with CS, CZ and CZ+CS, respectively.

Figure 6 shows the results for the levels of phosphorylated NOS (pNOS) (panel A). In control conditions, pNOS tended to decrease after two hours of perfusion and during ischemia. There is a significant decrease in the relative density of pNOS in the ischemia-reperfusion condition. Treatments with CS or CZ did not show significant changes. Surprisingly, treatment with CZ+CS shows an increase in the relative density of pNOS during ischemia and during reperfusion.

Panel B shows that the levels of iNOS were not modified in the hearts of animals without treatment and under perfusion conditions for 30 and 120 min, and during ischemia and reperfusion. Treatment with CS (panel B-2) decreased the levels of iNOS at 30 and 120 min of perfusion, with respect to the hearts of animals without treatment (B-1). The levels of iNOS increased significantly during ischemia and during reperfusion with respect to its control (perfusion 30 min) with the treatment with CS. Treatment with CZ (panel B-3) did not generate changes in the levels of iNOS, and the levels resembled those observed in the hearts of animals without treatment. With the combined treatment CZ+CS (panel B-4), the levels of iNOS were significantly increased during ischemia and decreased with reperfusion.

### 4.7. Histological Analysis

The histological sections in the C group did not show important modifications in the conformation of the cardiac tissue. Contraction bands are arranged in compact bundles of myocytes, separated by fibrous bands. It is possible to distinguish focally intercalated disks and ovoid nuclei as part of the normal histologic aspect of the cardiac muscle. However, a tendency to damage was observed in the tissue with ischemia and reperfusion. This tendency to damage was prevented in the hearts from groups treated with CS and with CS+CZ. However, in the groups treated with CZ plus ischemia and reperfusion, the cardiac muscle showed slight changes in relation to the CS plus ischemia (*p* = 0.03) and CS+CZ plus reperfusion groups (*p* = 0.01, respectively). In the CZ plus ischemia group, a variable undulation in the bundles of myofibrils was observed, and these cells were narrower than those in the controls in the longitudinal section. In the CZ group during reperfusion, the damage is more severe with the presence of edema, loss of striations, breakage of the fibers and focally incipient necrosis. We did not find hemorrhage, hyper eosinophilia, inflammatory cells or polymorph nuclear leukocytes.

## 5. Discussion

The damage that occurs in the myocardium during ischemia is related to changes in temperature, increases in pH, decreases in mitochondrial ATP and retention of lactic acid. After ischemic damage occurs, cardioprotective pathways such as NO and protein kinase G (PKG) are activated [34]. Cardiac function may be restored and a decrease in the size of the infarct produced by AMI may be accomplished. This has been proven by the studies reported by Jennings et al. in 1960, (on cardiac damage due to ischemia and reperfusion) [35,36] and the studies by Murry et al. in 1986 (on pre-conditioning and post-ischemic conditioning) and the restoration seems to depend on the time of exposure to ischemia [8,37]. In this sense, TRPV1 may importantly participate in cardioprotection. Although its role has been studied in different preconditioning, postconditioning and remote preconditioning models [38,39,40], the mechanism of transmission of signals that are involved in cardioprotection have not yet been elucidated. Therefore, in this paper we tested the role of TRPV1 in the possible restoration of cardiac function in ischemia and reperfusion. We explored how TRPV1 participates in the regulation of mechanical work and the involvement of some components of the nitric oxide pathway. We also explored the role of TRPV1 on oxidative stress and report the changes in the expression of iNOS, eNOS, pNOS and TRPV1.

Preclinical studies have shown that dietary CS, an agonist of TRPV1, modulates processes such as adipocyte browning and the activation of metabolic modulators, including AMP protein kinase (AMPK), peroxisome proliferator-activated receptor α (PPARα), uncoupling protein 1 (UCP1), USP2 and glucagon-like peptide 1 (GLP-1) [41]. The efficacy of this agent is greater at low doses. Moreover, CS blocks damage by anoxia/reoxygenation in cardiomyocytes by regulating mitochondrial function, preventing the opening of the mitochondrial pore and the release of cytochrome-C. The reduction and control of reactive oxygen species (ROS) by this agent is also promoted under these conditions [18,42]. Therefore, we explored the participation of CS in cardioprotection from ischemia-reperfusion damage. We also used an antagonist of TRPV1, CZ and, as expected, it not only inhibited the actions of CS during reperfusion but also produced damage from the pre-ischemic period.

In this paper we found that after exposure to global heart ischemia for 30 min, the stunned heart does not return to its normal mechanical activity with reperfusion. (Figure 2) This allowed us to compare and demonstrate that the activation of TRPV1 with the administration of CS at the systemic level reversed the ischemic and reperfusion damage in the isolated and perfused heart. We found that the heart loses 40% of its mechanical activity with CZ. Treatment with CZ+CS also generated a significant decrease in cardiac mechanical activity. The affectation by both these treatments is added to ischemic and reperfusion damage. Therefore, CS actions were inhibited by CZ and part of the explanation of this phenomenon is that CZ inhibits the flow of Ca^2+^ and modifies NO levels. It is possible that the alteration in NO levels and Ca^2+^ flux is decisive to decrease the mechanical activity.

We found an increase in the levels of NO, BH4 (Figure 3A,B) and CGRP (Figure 4B) in the heart due to the activation of TRPV1 before ischemia-reperfusion, and we suggest that these factors form part of a myocardial protection pathway. In addition, these variables were not modified in ischemia and reperfusion and adequate levels of these factors were maintained. A reflection of this improved state is that the TAC was modified accordingly (Figure 4A).

Regarding the role of TRPV1 during ischemia, we found that its expression was significantly increased in this condition (Figure 7). Therefore, the activation of TRPV1 might possibly generate a preventive state that could correspond to myocardial protection, thus eliminating ischemic and reperfusion damage. Therefore, the question arises whether CS may generate this cardioprotective state, and whether the mechanism for this protection is relevant to explore. One possibility is that the activation of TRPV1 might improve Ca^2+^ regulation, which is one of its well-known functional characteristics. It is also possible that it could induce a mitochondrial protective state, since the mitochondria are involved as effector organelles in cardioprotection [8].

Since the expression of TRPV1 and the levels of CGRP in cardiac tissue increased during ischemia and during reperfusion while PDE-3 and MDA decreased, we propose that CGRP may be one of the mediators of cardioprotection. The fact that TRPV1 is expressed during ischemia allows us to speculate that TRPV1 inhibits the opening of the mitochondrial transition pore and regulates Ca^2+^ flux, which leads to its possible participation in the control of the cell damage pathways that are activated during ischemia [43]. We are therefore currently working on the design of a study at this level. Although the TRPV1 protein has been mainly visualized in neuronal cells, there are studies that show its presence in the heart of the mouse. Furthermore, TRPV1 has been detected in mitochondrial fractions [16,18].

Due to an imbalance between the levels of NO and the cofactors required for its synthesis such as BH4, there is a change in the levels of cGMP and an increase in ROS. Under these conditions, the levels of biomarkers of cell damage increase. In this study we also analyzed the modifications in the levels of PDE-3 and MDA by the pharmacological activation or inhibition of TRPV1. The levels of both biomarkers increased in response to ischemia and reperfusion. However, treatment with CS decreased and controlled their levels. In other words, the activation of TRPV1 synchronized the NO pathway reactions that were altered by the ischemia and reperfusion events.

In our analysis, we did not find modifications in the expression of eNOS (Figure 5) with any of the treatments or with perfusion or ischemia and therefore, there was no effect on the constitutive state of the enzyme. However, when we analyzed the expression of pNOS, we found that the activation of TRPV1 decreased this form of the enzyme in hearts with each treatment, with respect to the hearts of animals without treatment. Interestingly, there is a tendency for a recovery of its relative density during reperfusion in the CS treated group.

Due to the modifications observed in the relative density of the iNOS in the CS group (Figure 6), we propose that the activation of TRPV1 regulates this enzyme, because the increases observed during ischemia and during reperfusion were not different from those observed in ischemia and reperfusion in the control groups, and with CZ or in the ischemia of the CZ+CS group. We do not have an explanation for the modification in the reperfusion of the CZ+CS group. It may be possible that the effect of CS is the predominant one when the combination treatment is administered. However, more observations in this regard are needed.

On the other hand, we were surprised that the relative density of TRPV1 increased with ischemia and reperfusion in the hearts of untreated animals. It is possible that this is part of a preventive response mechanism for the damage that is being generated.

NO levels in the heart are altered due to disease or to ischemia and reperfusion as found in our study. Oxidative stress is increased, and damage signals are activated. This also occurs even with the application of drugs in high doses in healthy animals. An imbalance in NO levels reduces its bioavailability because it rapidly interacts with ROS, causing an enzymatic uncoupling between eNOS and BH4, therefore inhibiting the formation cGMP. Activation of TRPV1 with CS maintains the increased levels of cGMP during ischemia. It is possible that the increased cGMP level acts as part of the mechanism for preventing damage and in the reparation processes. In reperfusion, TRPV1 levels return to normal values because this receptor is regulated by stimuli from the environment such as shear stress, changes in temperature and pH that occur during reperfusion. The regulation of the NO pathway is restored with CS and the damaged pathways are repaired.

With respect to the histological studies of cardiac tissue (Figure 8), we did not find important changes. Therefore, 30 min of global ischemia may be a short time to cause histological changes. However, we were able to distinguish myocyte damage and fiber breakage that were diminished by CS. The morphological changes correlate with cardiac work, since with ischemia and reperfusion damage, the mechanical activity of the heart is affected by only 20 or 25%. There were also variations in NO levels that did not affect the mechanical activity of the heart or TAC, as can be observed in the hearts of healthy animals with CS treatment that were perfused for one hour.

Therefore, the activation of TRPV1 may protect the heart by regulating oxidative stress, the NO pathway, and by controlling the flow of Ca^2+^ to prevent damage to the cardiac tissue when the proper functioning of the heart is compromised, as in ischemia and reperfusion. With this evidence, the possibility exists that TRPV1 is related to the mechanisms necessary to keep mitochondrial function in synchrony.

## 6. Conclusions

The activation of TRPV1 participates in the correction and control of alterations in the levels of NO, cGMP, BH4 and CGRP caused by ischemia reperfusion damage. The damage is manifested by the modifications to the mechanical activity of the heart. TRPV1 also controls the levels of MDA and PDE-3 in cardiac tissue. Therefore, we conclude that TRPV1 is involved in the regulation of the NO pathway, and in the production of reactive oxygen species, through a mechanism mediated by CGRP.

## Figures and Tables

**Figure 1 molecules-27-01031-f001:**
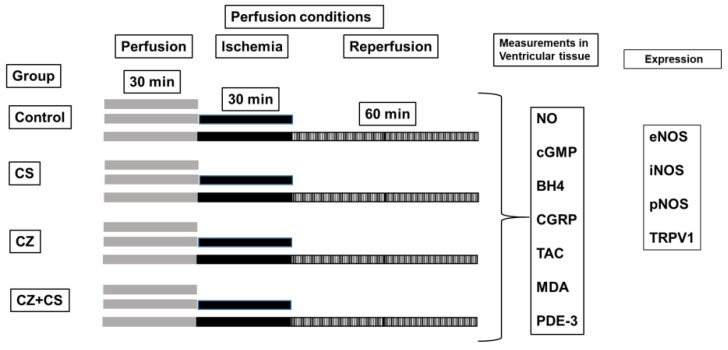
Experimental conditions. Each experimental group (Control, CS, CZ and CZ+CS) was subjected to different conditions: 1.—Pe (perfusion) 30 min. 2.—Pe 30 min + I (Ischemia) 30 min, without reperfusion (R). 3.—Pe 30 min + I (30 min) + Reperfusion for 60 min. Global ischemia was induced by stopping coronary flow by turning off the perfusion pump. Left ventricular tissue samples were obtained from all experimental conditions and frozen at −70 °C until metabolite detection tests were performed.

**Figure 2 molecules-27-01031-f002:**
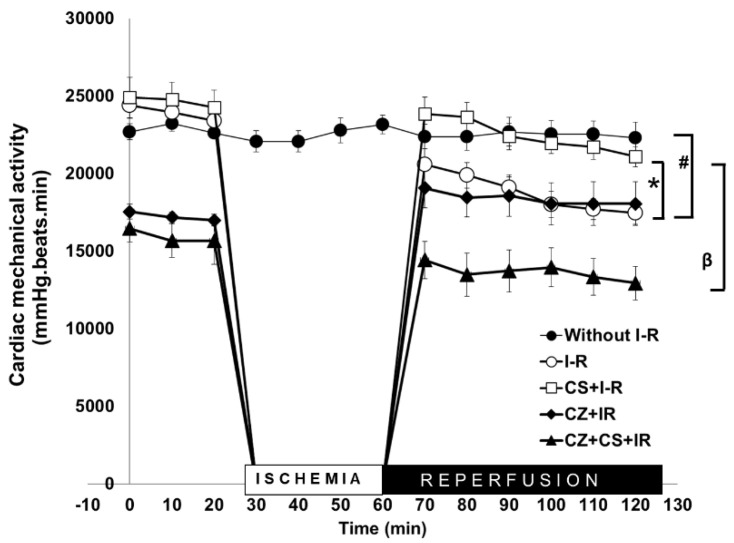
Cardiac mechanical activity in isolated and perfused hearts with perfusion (30 min), global ischemia (30 min) and reperfusion (60 min) periods for rats without treatment and after treatment with CS, CZ and combination CZ+CS. Values are the mean ± SE (*n* = 10 in each group) *p* ≤ 0.05 * CS + I-R vs. I-R; # Without I-R vs. I-R; β CS + I-R vs. CZ+CS + I-R.

**Figure 3 molecules-27-01031-f003:**
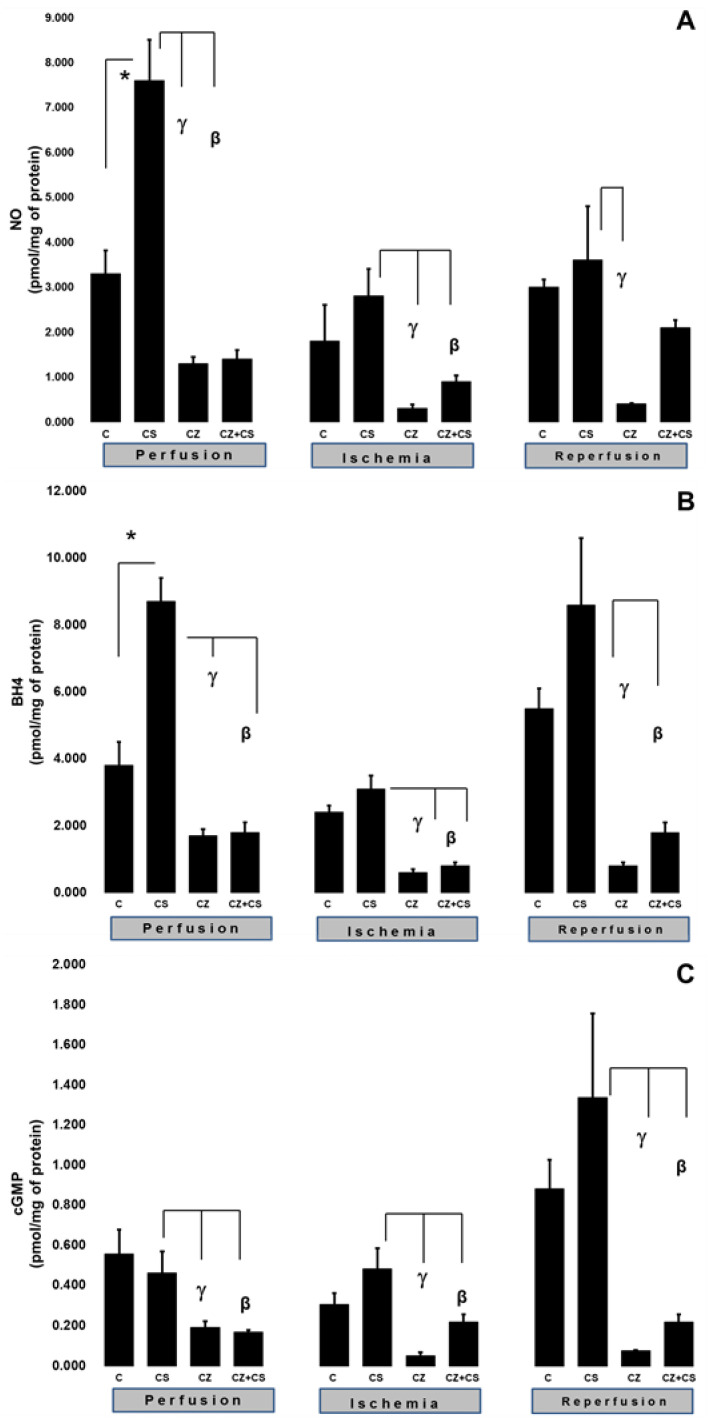
Levels in cardiac tissue of NO (**A**), BH4 (**B**) and cGMP (**C**). Isolated hearts were perfused in three conditions: perfusion (30 min), ischemia (30 min) and reperfusion (60 min). Hearts were obtained from control rats and treated with CS, CZ and CZ+CS. Ventricular tissue samples were obtained at the end of reperfusion. Values are the mean ± SE (*n* = 5 in each group *p* ≤ 0.05).* C vs. CS; γCS vs. CZ; βCS vs. CZ+CS.

**Figure 4 molecules-27-01031-f004:**
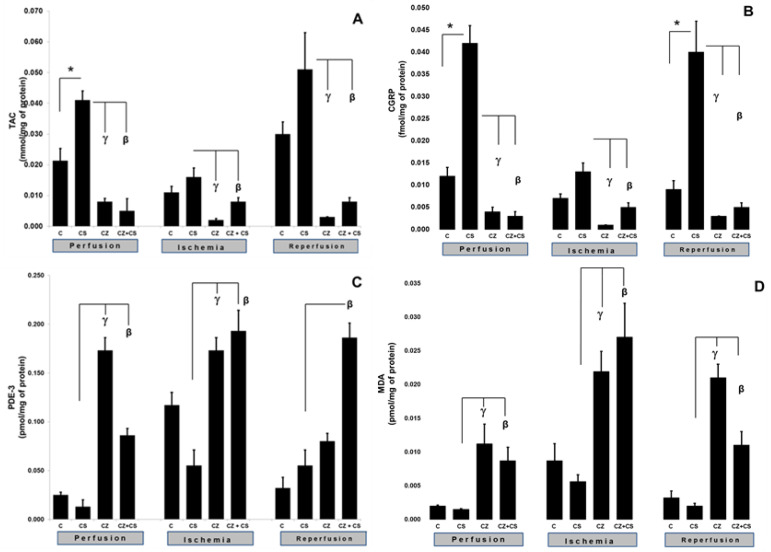
Cardiac TAC levels as an indicator of modifications in oxidative stress (**A**) and CGRP levels as an indicator of TRPV1 activation (**B**). Levels of tissue damage indicators PDE-3 (**C**) and MDA (**D**). The determinations of these biomarkers were made in ventricular tissue of isolated hearts perfused in three conditions: perfusion (30 min), ischemia (30 min) and reperfusion (60 min). Hearts were obtained from control rats and treated with CS, CZ and CZ+CS. Ventricular tissue samples were obtained at the end of reperfusion. Values are the mean ± SE (*n* = 5 in each group *p* ≤ 0.05).* C vs. CS; γ CS vs. CZ; β CS vs. CZ+CS.

**Figure 5 molecules-27-01031-f005:**
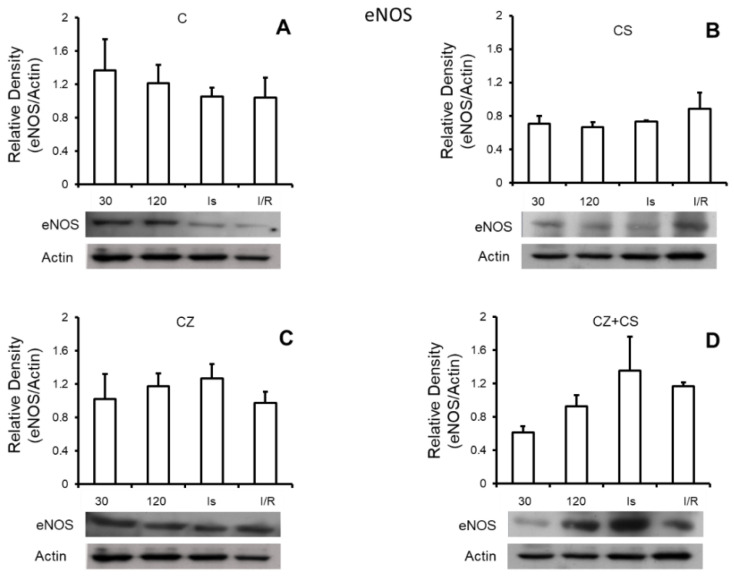
eNOS expression in isolated hearts from control (**A**) and treated animals with CS (**B**), CZ (**C**) and CZ+CS (**D**) and exposed to different perfusion times (30 min and 120 min), ischemia (Is) (30 min) and ischemia-reperfusion (I/R)(60 min). Values are the mean ± SE (*n* = 5 in each group *p* ≤ 0.05).

**Figure 6 molecules-27-01031-f006:**
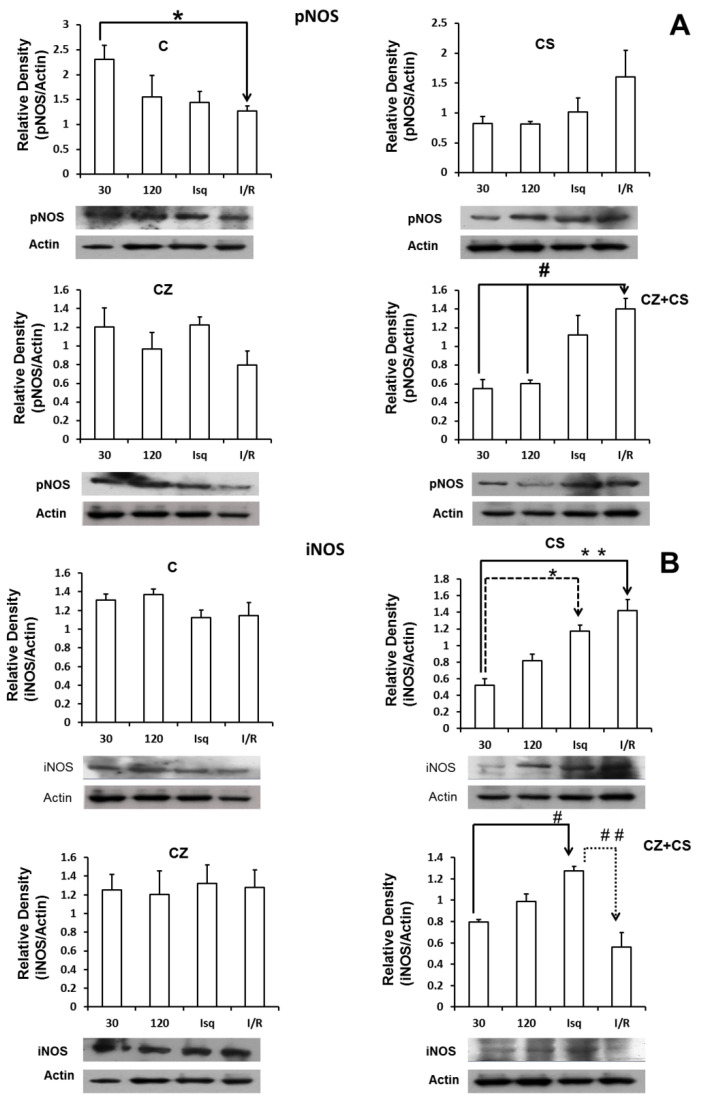
pNOS (**A**) and iNOS (**B**) expression in isolated hearts of rats treated with CS, CZ and CZ+CS under conditions of perfusion (30 and 120 min), ischemia (30 min) and reperfusion (60 min). Values are the mean ± SE (*n* = 5 in each group *p* ≤ 0.05). (**A**) * C 30min vs. I/R; ^#^ CZ+CS (30min vs. I/R). (**B**) * CS (30min vs. Is); ** CS (30min vs. I/R); ^#^ CZ+CS (30 min vs. Is); ^##^ CZ+CS (Is vs. I/R).

**Figure 7 molecules-27-01031-f007:**
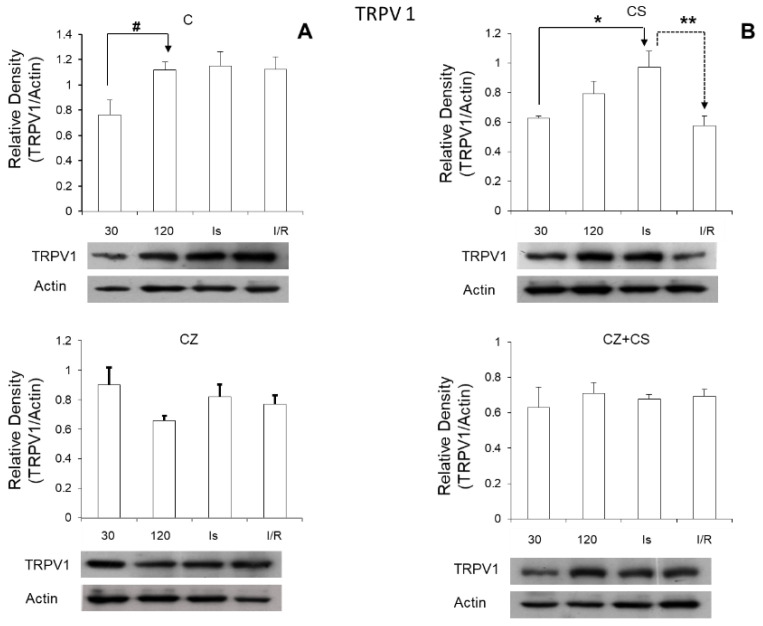
Levels of TRPV1 increased with the perfusion time (120 min) and with ischemia and reperfusion in the control group (**A**). Treatment with CS significantly increased the levels of TRPV1 during ischemia and the level remained at the baseline during reperfusion (**B**). Treatments with CZ and with the combination CZ+CS did not show changes in the levels of TRPV1 Values are the mean ± SE (*n* = 5 in each group *p* ≤ 0.05). # C (30 min vs. 120 min); * CS (30 min vs. ischemia); ** CS (ischemia vs. I/R).

**Figure 8 molecules-27-01031-f008:**
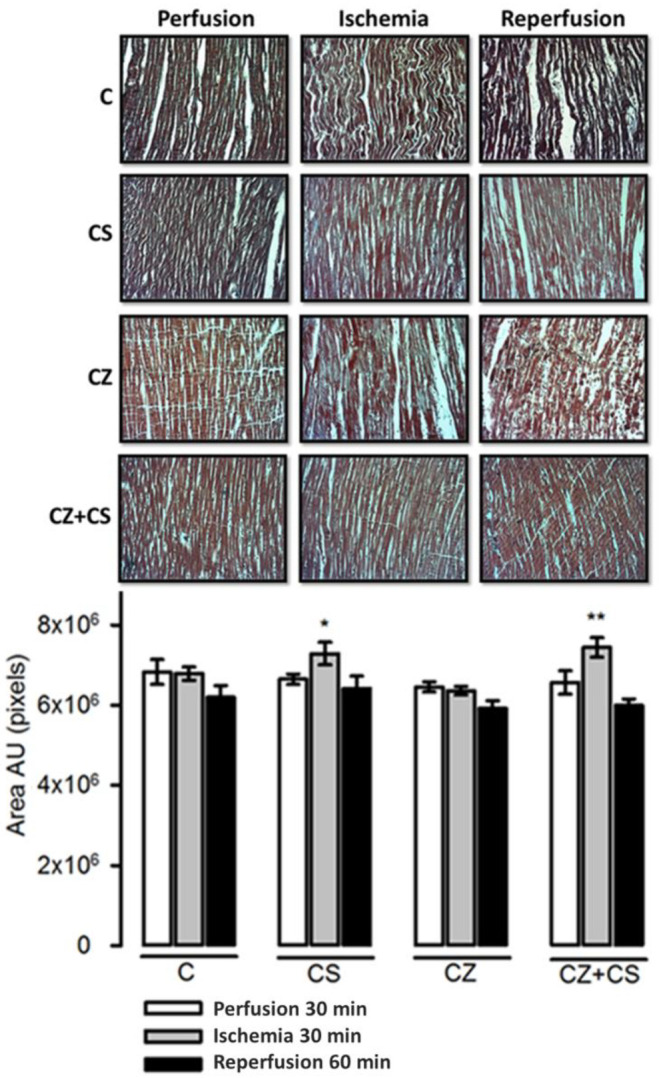
Representative photomicrographs of heart tissue after perfusion (30 min), ischemia (30 min) and reperfusion (60 min) from the four experimental groups. Area values are the mean ± SE (*n* = 5 in each group). The tissue was processed according to conventional histological procedures, and histological sections were made and stained by Masson’s trichrome stain at 16x. Control hearts with reperfusion showed injured in the myocytes with wavy fiber, but the CS treatment restored the morphology of the tissue. However, the CZ treatment was not capable of restoring morphology and breakage of the fibers was observed. The morphology of the tissue in the CZ+CS group was very similar to that of the C group. * CS + Ischemia (I) vs. CZ + I, *p* = 0.03; ** CZ + I vs. CS+CZ + I, *p* = 0.01.

## Data Availability

The data used to support the findings of this study are included within the article.

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
