# Peer review of "TRPV1 Contributes to Modulate the Nitric Oxide Pathway and Oxidative Stress in the Isolated and Perfused Rat Heart during Ischemia and Reperfusion"

_molecules, 2022, doi:10.3390/molecules27031031_

Round 1
Reviewer 1 Report
The present study by Castrejón Téllez et al. investigates the effects of TRPV1 receptors on the oxidative stress and nitric oxide pathway during ischemia/reperfusion injury in isolated rat hearts. The study is basically interesting, however, the concept of the study is unclear, the rational of the measured parameters is not obvious and it is not explained by the authors. Moreover, the interpretation of the data as well as the English editing require substantial improvements. I would suggest some major issues to be improved:
- The authors are advised to ask assistance of a native English speaker for improving language editing of the manuscript
- The title of the manuscript can be shortened and “potential” should be written consistently: either with a capital initial letter or capital initial letters can be ignored from the full name of TRPV1.
- A protocol figure should be included to improve understanding of the study design.
- The concept of the study is unclear. The authors do not explain thoroughly the rational of the measured factors/parameters. Why they did choose just that particular biomarkers is also unclear. Please, provide more details about the concept of the study. Using regional ischemia even in isolated hearts has higher potential to be translated to human conditions than that of global ischemia, which is clinically irrelevant.
- The quality of most of the original Western blot images are poor. The legends for all figures should be extended, especially abbreviations should be explained. Figure 1 contains redundant diagrams, the data of panels A, B and C can be presented as one diagram. What was the statistical analysis used for the evaluation of cardiac work data? It is not described even in the statistical analysis section. The definition of cardiac work is the product of stroke work and heart rate, which is the equivalent of the triple product of stroke volume, aortic pressure and heart rate. Since the first two parameters were not determined, instead of cardiac work the use of rate pressure product is rather suggested, which is the product of arterial (in this particular study, ventricular) pressure and heart rate.
- Materials and methods section is also very brief and poorly written. Some of the used methods are described thoroughly (e.g. Langendorff perfusion and PDE3 assay), however, most of them need to be extended and described in more detail. I do not see the sense of centrifugation at 10°C – is there any specific reason for not using the conventional 4°C? Please clarify.
- Please cite the most recent review paper (Int J Mol Sci. 2020;21(12):4472. doi: 10.3390/ijms21124472) discussing very close topics to this manuscript (including the role of TRPV1 channels in cardiac ischemia/reperfusion injury) in the introduction and discussion sections of this manuscript. Another research paper should be cited investigating the role of capsaicin sensitive sensory nerves in cardiac dysfunction and oxidative stress (Br J Pharmacol. 2008;153(3):488-96. doi: 10.1038/sj.bjp.0707599)
Author Response
Response to Reviewer 1 Comments
Point 1: The authors are advised to ask assistance of a native English speaker for improving language editing of the manuscript.
Response 1: The paper was reviewed by a native english speaking researcher and changes in the language were made throughout the manuscript.
Point 2: The title of the manuscript can be shortened and “potential” should be written consistently: either with a capital initial letter or capital initial letters can be ignored from the full name of TRPV1.
Response 2: We have reduced the title by putting the name TRPV1, which is how the capsaicin receptor is known and thus making it more suitable for reading.
“TRPV 1 receptor contributes to modulate the nitric oxide pathway and oxidative stress in the heart of rat during ischemia and reperfusion”.
Point 3: A protocol figure should be included to improve understanding of the study design.
Response 3: Figure 1, which corresponds to the experimental protocol that we carried out in this investigation, was added to the document.
Point 4: The concept of the study is unclear. The authors do not explain thoroughly the rational of the measured factors/parameters. Why they did choose just that particular biomarkers is also unclear. Please, provide more details about the concept of the study. Using regional ischemia even in isolated hearts has higher potential to be translated to human conditions than that of global ischemia, which is clinically irrelevant.
Response 4: The biomarkers studied are overexpressed or inhibited by various biological processes related to oxidative stress and cell damage, such as MDA, which is the product of lipoperoxidation, increasing cell damage and death, and PDE-3, which alters the pathway of cGMP inhibiting relaxation.
CGRP, BH4, NO and cGMP are indicators of alterations in vasorelaxation, endothelial damage and diseases such as arterial hypertension.There is a decrease in serum CGRP levels and a decrease in the expression of eNOS attributed to enzymatic uncoupling by BH4, and in these conditions NO levels are altered. These biomarkers are sensitivity to changes in the vascular environment or to diseases and it is possible to study them. Moreover, previous studies suggest that TRPV1 is a regulator of the cGMP-NO pathway.
Regarding global ischemia, you are right because it has no clinical relevance. Perhaps the closest thing to an experimental study of this type would be to analyze the biochemical modifications and the response of the heart to a preconditioning (in this case with capsaicin) for a transplant or surgery, but this is just an idea that requires further studies. Therefore, we are designing a research protocol with a regional ischemia generated by a ligation of the descending coronary artery. We want to direct the study to CGRP as a molecule with cardioprotective properties which will be the subject of a future paper.
Point 5: The quality of most of the original Western blot images are poor. The legends for all figures should be extended, especially abbreviations should be explained. Figure 1 contains redundant diagrams, the data of panels A, B and C can be presented as one diagram. What was the statistical analysis used for the evaluation of cardiac work data? It is not described even in the statistical analysis section. The definition of cardiac work is the product of stroke work and heart rate, which is the equivalent of the triple product of stroke volume, aortic pressure and heart rate. Since the first two parameters were not determined, instead of cardiac work the use of rate pressure product is rather suggested, which is the product of arterial (in this particular study, ventricular) pressure and heart rate.
Response 5: We made a new image of the Western blot with the maximum resolution of our equipment and we achieved few modifications. Figure 1, which corresponded to the results obtained in the isolated heart, was modified in a single panel. Now it is figure 2 since figure 1 corresponds to the protocol.
The statistical analysis used for cardiac mechanical activity was a one-way analysis of variance (ANOVA) followed by a Tukey post hoc test using the SigmaPlot program version 11.
The product of left ventricular pressure and heart rate were used as hemodynamic parameters to calculate the mechanical activity of isolated hearts. Your observation is very important and we accept our error in expressing that it is cardiac work when in reality we are working with the isolated heart, which has limitations such as retrograde perfusion and the absence of fluid flow through the ventricles. Therefore, we refer to the product of left ventricular pressure and heart rate as cardiac mechanical activity.
Point 6: Materials and methods section is also very brief and poorly written. Some of the used methods are described thoroughly (e.g. Langendorff perfusion and PDE3 assay), however, most of them need to be extended and described in more detail. I do not see the sense of centrifugation at 10°C – is there any specific reason for not using the conventional 4°C? Please clarify.
Response 6: The NO and TAC methodologies cannot be expanded further, since they are spectrophotometric methods in the Visible region, which are simple and do not require to be detailed as in the case of the other biomarkers (BH4, cGMP, PDE-3 and CGRP). These other methodologies are more complex and require greater care. We therefore, expanded their explanation.
It is also true that within the methodology, the centrifugations were carried out at 10 °C because the biomolecules studied remain stable below 18 °C. Initially we worked with centrifugation at 4 °C but part of the precipitated material adhered to the wall of the tubes and therefore we changed to an intermediate temperature and obtained better results. This is the only reason why we used 10 °C.
Point 7: Please cite the most recent review paper (Int J Mol Sci. 2020;21(12):4472. doi: 10.3390/ijms21124472) discussing very close topics to this manuscript (including the role of TRPV1 channels in cardiac ischemia/reperfusion injury) in the introduction and discussion sections of this manuscript. Another research paper should be cited investigating the role of capsaicin sensitive sensory nerves in cardiac dysfunction and oxidative stress (Br J Pharmacol. 2008;153(3):488-96. doi: 10.1038/sj.bjp.0707599)
Response 7: Thank you very much for suggesting the review by Szabados et al., We have read it and included it in the text and in the references. It is an excellent review work and opens the possibility to design and conduct new research involving the functionality of the TRPV1. As mentioned in this review, non-neuronal TRPV1 is found in various cell types and we found of interest that it is present in the mitochondria, as we mentioned in the discussion of our work.
We appreciate your comments on our work and we thank you in advance for your attention.

Reviewer 2 Report
Manuscript ID: molecules-1523596
Manuscript title: Transient Receptor potential Vanilloid type 1 contributes to modulate the nitric oxide pathway and oxidative stress in the isolated and perfused rat heart during ischemia and reperfusion.
Authors investigated the effects of capsaicin (CS) and/or capsazepine (CZ) on nitric oxide and oxidative stress markers in the isolated and perfused hearts during ischemia/reperfusion. The manuscript is interesting, but there are a few concerns to consider for publication.
In Figure 7, authors demonstrated the microphotographs of hematoxylin & Eosin staining and they demonstrated the tendency to damage was prevented in the groups treated with CS and with CS+CZ. However, I could not observe any remarkable differences on the morphology. Authors should conduct Masson’s trichrome staining and TTC staining to show the damage areas to support the biochemical data.
Authors should confirm the side effects of CS and/or CZ such as denervation in nerve ending.
Author Response
Response to Reviewer 2 Comments
Authors investigated the effects of capsaicin (CS) and/or capsazepine (CZ) on nitric oxide and oxidative stress markers in the isolated and perfused hearts during ischemia/reperfusion. The manuscript is interesting, but there are a few concerns to consider for publication.
We reviewed the methods and results section, we made modifications in the manuscript for a better reading and transmission of procedures.
We have included a reference that reinforces the concepts mentioned in the work (Szabados et al. 2020).
Point 1: Manuscript title: Transient Receptor potential Vanilloid type 1 contributes to modulate the nitric oxide pathway and oxidative stress in the isolated and perfused rat heart during ischemia and reperfusion.
Response 1: The paper was reviewed by a native english speaking researcher and changes in the language were made throughout the manuscript.
We have reduced the title by putting the name TRPV1, which is how the capsaicin receptor is known and thus making it more suitable for reading.
TRPV 1 receptor contributes to modulate the nitric oxide pathway and oxidative stress in the heart of rat during ischemia and reperfusion.
Point 2: In Figure 7, authors demonstrated the microphotographs of hematoxylin & Eosin staining and they demonstrated the tendency to damage was prevented in the groups treated with CS and with CS+CZ. However, I could not observe any remarkable differences on the morphology. Authors should conduct Masson’s trichrome staining and TTC staining to show the damage areas to support the biochemical data.
Response 2: Looking at figure 7, you are right. As part of the methodology, every time we do tissue marking we always have two options such as marking with Hematoxylin and Eosin as well as Masson's trichrome. We substituted the image stained with HE for the image stained with Masson`s in figure 8. However, we did not find important differences with either of the two options, but with hematoxylin and eosin we found slight modification during ischemia in the groups with CS and with CZ+CS that we attribute to a state of prevention. This is because the exposure of the heart to ischemia was 30 min and it is a very short time to obtain important structural changes. From the onset of ischemia and as time progresses, the chances of restoring the mechanical activity of the heart to 100% with reperfusion decrease. In our study with 30 min of ischemia and 60 min of reperfusion, the heart recovered approximately 80%, the tissue deterioration is in an initial period and the deficiencies in the mechanical activity of the heart are due to changes at the biochemical level such as the decrease in the bioavailability of NO and the increase in oxidative stress. In the new figure (Masson's trichrome) it is possible to observe areas of fiber rupture and damage to myocytes in groups C, CZ. That damage is not seen in the CS group.
Point 3: Authors should confirm the side effects of CS and/or CZ such as denervation in nerve ending.
Response 3: In this work we did not analyze the denervation effects that capsaicin can produce because for this to occur the CS concentration must be high (> 50 mg/kg) and in our study we used 20 mg/kg. We did not see any evidence of neuronal damage. The application of capsaicin should be subcutaneous so that the absorption is gradual and there are no toxic effects.
We appreciate your comments on our work and we thank you in advance for your attention.

Round 2
Reviewer 2 Report
The manuscript has been improved and I have no further comments.
Author Response
REVIEWER 2
Point 1: English language and style are fine/minor spell check required.
Response: Review of English was also carried out by a researcher whose native language is English.
